# Histone H2AX deficiency causes neurobehavioral deficits and impaired redox homeostasis

Urbain Weyemi[1], Bindu D. Paul[1], Adele M. Snowman[1], Parthav Jailwala[2], Andre Nussenzweig[3], William M. Bonner[4] & Solomon H. Snyder[1,5,6]

ATM drives DNA repair by phosphorylating the histone variant H2AX. While ATM mutations elicit prominent neurobehavioral phenotypes, neural roles for H2AX have been elusive. We report impaired motor learning and balance in H2AX-deficient mice. Mitigation of reactive oxygen species (ROS) with N-acetylcysteine (NAC) reverses the behavioral deficits. Mouse embryonic fibroblasts deficient for H2AX exhibit increased ROS production and failure to activate the antioxidant response pathway controlled by the transcription factor NRF2. The NRF2 targets GCLC and NQO1 are depleted in the striatum of H2AX knockouts, one of the regions most vulnerable to ROS-mediated damage. These findings establish a role for ROS in the behavioral deficits of H2AX knockout mice and reveal a physiologic function of H2AX in mediating influences of oxidative stress on NRF2-transcriptional targets and behavior.

[1] The Solomon H. Snyder Department of Neuroscience, Johns Hopkins University School of Medicine, Baltimore, MD 21205, USA. [2] Center for Cancer Research Collaborative Bioinformatics Resource, National Cancer Institute, Bethesda, MD 20892, USA. [3] Laboratory of Genome Integrity, National Cancer Institute NIH, Bethesda, MD 20892, USA. [4] Developmental Therapeutics Branch, Laboratory of Molecular Pharmacology, National Cancer Institute, National Institutes of Health, Bethesda, MD 20892, USA. [5] Department of Psychiatry and Behavioral Sciences, Johns Hopkins University School of Medicine, Baltimore, MD 21205, USA. [6] Department of Pharmacology and Molecular Sciences, Johns Hopkins University School of Medicine, Baltimore, MD 21205, USA. Correspondence and requests for materials should be addressed to S.H.S. (email: ssnyder@jhmi.edu)

Maintaining genome stability is fundamental to the development and survival of any organism. Mammalian cells are endowed with a number of tightly regulated pathways providing efficient DNA repair in responses to endogenous or exogenous genotoxic agents. Compromising genomic DNA can lead to a broad range of human diseases including immune deficiency, cancer, age-related diseases, and neurodevelopmental disorders. DNA repair deficiency in the brain can lead to microcephaly, neurodegeneration or brain tumors[1].

The functional unit of chromatin is the nucleosome. It contains about 147 bp of DNA wrapped around an octamer of histones composed of an (H3-H4)$_2$ tetramer flanked by two dimers of H2A-H2B[2]. These four nucleosome histone families provide equal numbers of molecules to the nucleosome. However, several of the families include members with variant sequences, whose stoichiometry can vary with cell type and growth state[3, 4]. The histone variant H2AX belongs to the histone H2A family, and like other histone variants, it is highly conserved and performs critical cellular functions beyond those fulfilled by the canonical H2As[5]. H2AX plays essential roles in DNA double-strand break repair and genome stability, and is classified as a tumor suppressor[6]. Within minutes following DNA double-strand breaks (DSBs), a cascade of signaling factors triggers the recruitment of hundreds of molecules essential for the repair and the resolution of the breaks. Repair of DNA DSBs, the most lethal reported DNA breaks, occurs through non-homologous end joining (NHEJ) or homologous recombination (HR)[6]. HR, an error-free process, preferentially takes place in proliferating cells like progenitors, whereas NHEJ, an error-prone process, operates both in proliferating, as well as differentiated cells such as neurons[7]. One of the early events in the DNA repair process is phosphorylation of the histone H2AX on its Ser139 residue (also called γH2AX)[8]. Upon induction of DSBs, DNA free ends recruit the MRN (MRE11–RAD50–NBS1) complex, leading to autophosphorylation of ATM (ataxia-telangiectasia mutated) kinase, which in turn induces formation of γH2AX. Additional kinases mediating DNA repair include ATR and DNA-PK. γH2AX creates binding sites for the MDC1 protein that serves as a docking site for additional DSB-repair proteins, facilitating the execution of the DNA damage response program[6]. Deficiency in the DNA repair kinase ATM leads to a fatal autosomal recessive neurological disease known as ataxia telangiectasia (AT). The frequency of AT in the United States is between 1:40,000 and 1:100,000, resulting in a carrier frequency of 0.5–1.0%[9]. AT patients exhibit a wide range of clinical manifestations including progressive cerebellar degeneration, oculomotor dysfunction, increased incidence of cancer, endocrine abnormalities, choreoathetosis, immunodeficiency, growth retardation, incomplete sexual maturation, and premature aging of the skin and hair[10]. ATM-null mice provide a model for phenotypes observed in patients with AT. However, ATM-null mice die from lymphomas before the onset of cerebellar degeneration[11]. These mice exhibit oxidative stress, as well as impaired general locomotor activity and reduced motor coordination[12, 11]. While ATM is best known for its function in DNA repair, it also mediates cellular reactive oxygen species (ROS) homeostasis[13]. While ATM serine 1981 is phosphorylated in response to DNA breaks, a cysteine residue located at the C-terminus (Cys 2991) participates in a homodimer formation during oxidative stress[13]. Chen et al. recently depicted the role of ATM in redox homeostasis, showing that ATM activation by oxidative stress leads to a transcriptional program that includes IL-8, during progression of primary tumors into metastatic nodules[14]. Several pathological manifestations of ATM deficiency can be reversed by the addition of antioxidants, confirming the relevance of ATM in regulating redox homeostasis[15–19]. Similarities between ATM and the histone variant H2AX are evident in the phenotypes of their knockout mouse models. In both instances males are sterile, and there is genomic instability evidenced by abnormalities in chromosome structure, immunodeficiency, and enhanced radiosensitivity. These similarities most likely reflect their common roles in DNA repair. However, while ATM mutants display prominent neurobehavioral phenotypes, neural roles for H2AX have been elusive. Here, we show that the loss of H2AX leads to neurobehavioral deficits. H2AX mediates physiologic responses to oxidative stress through NRF2-transcriptional targets, and antioxidant treatment ameliorates the neurobehavioral deficits of H2AX mutants.

## Results

**H2AX loss impairs motor balance and locomotor activity.** Ataxia Telangiectasia patients display defects in motor activity associated with cerebellar abnormalities including diminished Purkinje and granule cell populations. Oxidative stress also plays a role in disease progression, as the severity of AT is reduced by decreasing levels of ROS[15, 19, 20]. Among other actions, ATM phosphorylates the histone variant H2AX[6, 21]. Deletion of H2AX recapitulates features of ATM deletion including defective DNA repair, chromosomal abnormalities, immunodeficiency, and an increased risk of lymphoma[22, 23]. Combined germline inactivation of ATM and H2AX leads to early embryonic lethality, indicating that the two genes possess non-redundant functions in embryonic cells[24]. Whether H2AX impacts oxidative stress, the central nervous system, and related ATM functions is unclear.

Utilizing mice deleted for H2AX, we have defined functions for this histone in motor activity and balance (Fig. 1). In rotarod models, younger wild-type and mutant mice displayed similar baseline motor coordination, whereas in 4–5-month old animals the latency for H2AX mutants to fall was about half that of wild type. Wild-type mice improved their performance roughly 2-fold over four trials, but the mutants displayed no notable improvement (Fig. 1a). The impaired rotarod performance of the mutants was not associated with lack of strength, as the knockout mice were able to remain suspended upside down from a wire lid for the same duration as wild type.

We monitored spontaneous locomotor activity in an open-field model (Fig. 1b, c). Mice were placed in the center of the open field, and their movement around the arena (horizontal activity), as well as their rearing behavior (vertical activity) was assessed over a 45 min period (Fig. 1b, c). Three to five month old H2AX mutants exhibited decreases of about 50 % in both horizontal and vertical activities, defects resembling those reported for ATM mutants[11].

**ROS mediate H2AX loss-induced neurobehavioral deficits.** A number of neurologic disorders associated with defective DNA repair involve enhanced oxidative stress and/or deficits in repair of oxidative DNA lesions[1, 25]. Earlier we showed that the antioxidant N-acetylcysteine (NAC) relieves behavioral abnormalities associated with Huntington's disease[26], and others observed similar beneficial effects of NAC in AT mice[16, 20]. When we administered 20 mM of NAC to H2AX knockout mice in their drinking water from weaning until 4 months of age, the abnormalities of horizontal (Fig. 2a) or vertical activity (Fig. 2b) in the open field were partially reversed with negligible effects on motor balance (Fig. 2c). Wild-type mice treated with NAC did not show noticeable difference in behavior (Supplementary Fig. 1). These observations are in line with previous findings showing a lack of NAC effects on control mice when comparing several abnormalities induced by ATM deficiency including genomic instabilities[16].

The beneficial effect of NAC implies some defects of the oxidative stress response in the mutant mice. We monitored oxidative stress in mouse embryonic fibroblasts (MEFs) quantifying ROS with dihydroethidium (DHE). H2AX deletion was associated with a 60% increase in ROS levels (Fig. 2d, e). When stained with the mitochondrial marker MitoTracker, the mitochondria in the H2AX mutant MEFs appeared swollen and less filamentous than those in control samples (Fig. 2f, g). These abnormalities are reminiscent of cells with disturbed mitochondrial morphology and activity. H2AX mutant cells exhibited substantially increased levels of mitochondrial ROS as revealed by the MitoSOX Red stains (Fig. 2h, i). These data indicate a role for H2AX in ROS deposition and are consistent with previous findings reporting depletion of H2AX upon chronic exposure to

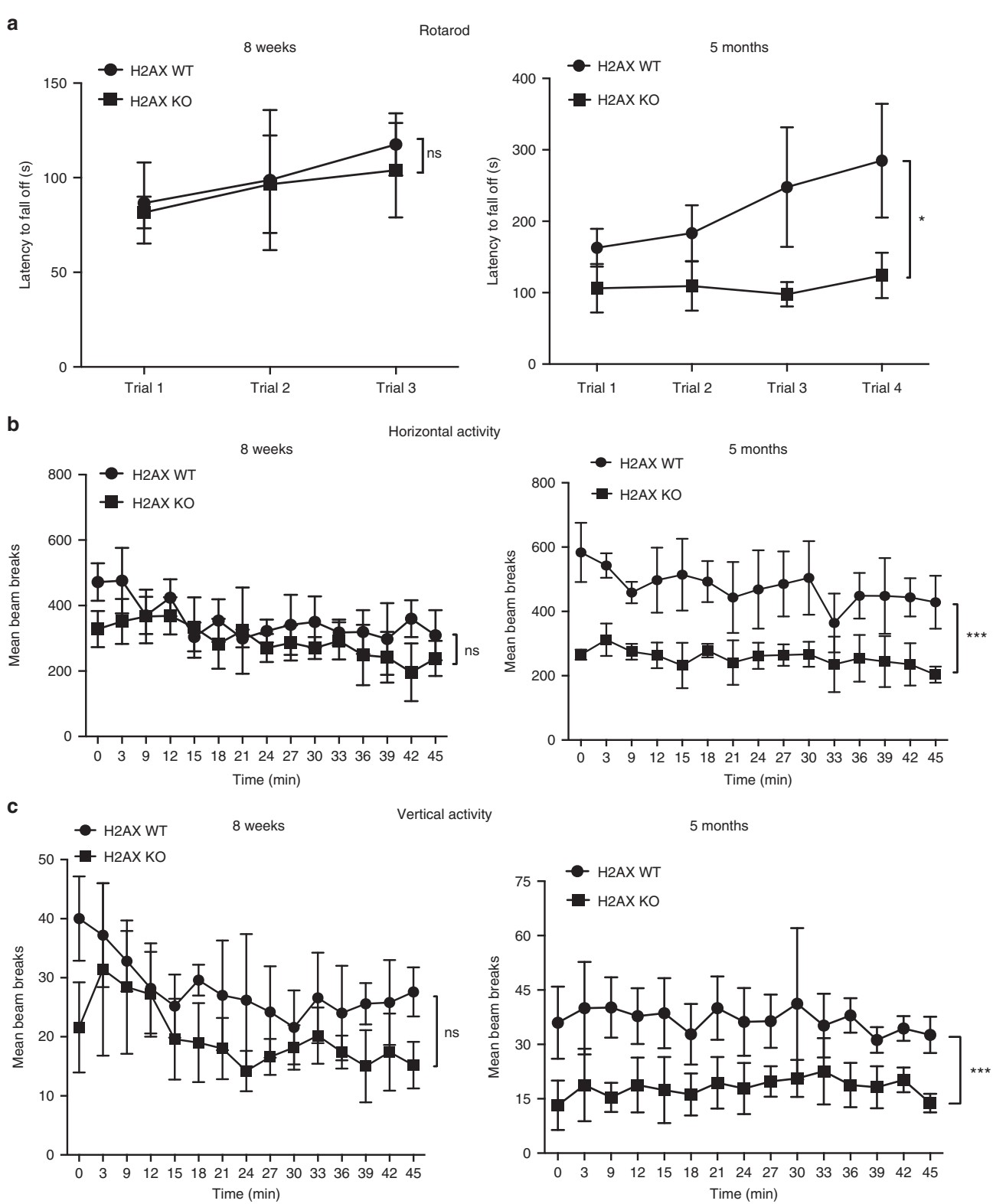

oxidative stress[27]. Considering the evidence that ATM phosphorylates H2AX in response to DNA damage, and that ATM deletion promotes accumulation of ROS, we explored the concomitant inactivation of ATM and H2AX. When treated with the ATM-specific inhibitor KU55933, H2AX-deficient cells produced about 40% more ROS, suggesting that the dual inactivation of H2AX and ATM is progressively more detrimental to a cell's redox balance (Supplementary Fig. 2). These observations fit with evidence that combined ATM/H2AX deficiency causes embryonic lethality and substantial genomic instability[28].

**H2AX deficiency impairs NRF2-regulated antioxidant response**. We examined the sensitivity of H2AX-deficient cells to ROS. MEFs were treated with $H_2O_2$ and monitored for long-term cell survival using a clonogenic assay (Fig. 3a). H2AX mutant cells were substantially more sensitive to the lethal effects of $H_2O_2$ than wild-type cells. As H2AX is an essential mediator of DNA repair[6], one might anticipate substantial sensitivity of H2AX-depleted cells to exogenous genotoxic oxidants like $H_2O_2$. To rule out this possibility cells were exposed to an endogenous ROS inducer, buthionine sulfoximine (BSO), a specific inhibitor of glutamate-cysteine ligase (GCLC), one of the key enzymes in the biosynthesis of glutathione. Ten micromolar of BSO reduced survival of wild-type cells about 25–30%. By contrast, survival of H2AX knockout cells was reduced about 80% (Fig. 3b), indicating an impaired antioxidant response. In summary, H2AX knockouts display increased sensitivity to oxidative stress.

To identify specific genes associated with oxidative metabolism that may be altered in H2AX mutants, we conducted a genome-wide differential gene expression analysis comparing H2AX-deficient and control MEFs. A heat map was generated by creating a list of genes involved in ROS metabolism from the 1295 differentially expressed genes, on the basis of their annotation in gene ontology. We identified 21 responsive genes of which NRF2 and its transcriptional target NQO1 appeared to be particularly relevant (Supplementary Fig. 3 and Supplementary Table 1). NRF2 is a transcription factor which mediates antioxidant responses[29]. Levels of NRF2 were reduced about 40% in the H2AX mutants, while NQO1 and GCLC, two major transcriptional targets of NRF2 that participate in detoxifying ROS, were reduced 75% and 35%, respectively (Fig. 3c, d). In addition, messenger RNA levels for GCLC, NRF2, and NQO1 were substantially lower in H2AX mutants (Fig. 3e). To further explore a role for NRF2, we measured the promoter activity of the antioxidant response element (ARE) binding site for NRF2, which was diminished by about 40% in the mutants (Fig. 3f).

To characterize the defective ROS response in H2AX knockout cells, we added BSO to their cultures to induce oxidative stress.

Levels of the NRF2 targets NQO1 and GCLCs in the mutants were markedly diminished (Fig. 4a, b). Thus, H2AX loss impairs the ability of cells to activate antioxidant response genes after exposure to oxidants. H2AX mutant cells rescued by H2AX re-expression (REV: revertants) displayed a pattern for GCLC and NQO1 similar to those of wild-type preparations (Fig. 4c, d). Real-time PCR (RT-PCR) confirmed the re-expression of NQO1 and GCLC in revertant cells, further substantiating evidence that the promoter activity of NRF2-binding site was impaired in mutant cells (Supplementary Fig. 4a, b). This shows that the deficits we observed in the H2AX mutants were not the result of other unrelated differences between the wild type and mutants. One intriguing question is whether the DNA repair-associated motif of H2AX, a phosphorylatable S139 residue, is required for control of redox homeostasis. To test this possibility, we generated H2AX knockout cells expressing a mutant H2AX (H2AX S139A) that is unable to detect DNA damage. Our results indicate that activation of NRF2-transcriptional targets NQO1 and GCLC was impaired in H2AX mutants (S139A), just as in the H2AX-null cells (Supplementary Fig. 4c-f). These findings indicate that the DNA repair function of H2AX is essential for its role in redox homeostasis. Moreover, as a direct target of ATM in the DNA damage response pathway, H2AX also seems to phenocopy ATM functions in the proper handling of oxidative stress. However, the role of H2AX seems related more to the activation of the antioxidant response than to the direct sensing of ROS. How S139 phosphorylation controls the direct transcriptional regulation of NRF2 targets still remains unclear.

To assess the influence of H2AX deletion in intact animals, we examined the striatum, which is notably vulnerable to ROS damage[26, 30]. Both NQO1 and GCLC were substantially diminished in H2AX mutants (Fig. 4e, f). To analyze whether the reduced expression of NQO1 and GCLC is consistent with elevated ROS damage in the striatum, we monitored levels of global protein carbonylation in the striatum and in primary neurons from both wild type and H2AX knockout mice using method described in the Supplementary Note 1. Proteins are among the major targets of ROS whose oxidative modification often alters enzymatic activity[31], DNA binding activities of transcription factors[32], and susceptibility to proteolytic degradation[33]. H2AX deletion elicited a 30% increase of protein oxidation in the striatum (Supplementary Fig. 5a, b) and more than a 2-fold increase in primary neurons (Supplementary Figure 5c). Similar results were obtained in MEFs in which H2AX loss led to a 40% increase in ROS-mediated protein damage (Supplementary Fig. 5d). These oxidative lesions were partially mitigated in the brain by NAC, consistent with alleviation of neurobehavioral deficits by antioxidant treatment (Supplementary Fig. 5a, b). The

**Fig. 1** H2AX loss leads to impaired motor balance and general locomotor activity. **a** Rotarod tests reveal reduced motor coordination in H2AX knockout mice. Testing was conducted over 3 days with 2 days of training trials. Each mouse was then given three to four trials and latency to fall off the rotarod was measured as it accelerated from 4 to 40 rpm over a 5 min period; y-axis, the latency to fall off in seconds, x-axis, number of trials. Each trial value relates to the average of the latency to fall off (in sec) for the total number of animals used per genotype. Statistical significance was determined by a two-tailed, unpaired Student's t-test; n = 7 (means ± s.e.m.) for H2AX wild type, n = 8 (means ± s.e.m.) for H2AX knockout; *P < 0.05, ns non-significant (P = 0.8570). Left panel represents data for 8-week-old mice; right panel summarizes data for 5-month-old mice. **b** Open-field testing demonstrated that wild-type mice were significantly more active than H2AX knockout mice as measured by horizontal activity; y-axis, beam breaks, x-axis, 3 min intervals. Left panel represents data for 8-week-old mice; right panel summarizes data for 5-month-old mice. Statistical significance was determined by a two-tailed, unpaired Student's t-test; n = 7 (means ± s.e.m.) for H2AX wild type, n = 8 (means ± s.e.m.) for H2AX knockout; ***P < 0.0001, ns non-significant (P = 0.1311). **c** Measurement of the rearing activity in open-field testing demonstrated that wild-type mice were rearing more actively compared with the mutant mice; y-axis, beam breaks, x-axis, 3 min intervals. Left panel represents data for 8-week-old mice; right panel summarizes data for 5-month-old mice. Statistical significance was determined by a two-tailed, unpaired Student's t-test; n = 7 (means ± s.e.m.) for H2AX wild type, n = 8 (means ± s.e.m.) for H2AX knockout; ***P < 0.0001, ns non-significant (P = 0.5413). The same mice used for the rotarod analyses are utilized for the open field tests following the same time points (8 weeks and 5 months)

elevated striatal protein damage paralleled our observations in H2AX-deficient mice of abnormal hindlimb clasping and clenching, characteristics which are reminiscent of mouse models with oxidative damage in areas such as the striatum (Supplementary Fig. 5e)[26]. Thus, multiple lines of evidence establish that H2AX deletion in mice interferes with NRF2-regulated antioxidant responses leading to oxidative stress and neurobehavioral deficits.

## Discussion

Defects in single genes classified as DNA repair genes often lead to severe neurologic disorders[25], but the mechanistic basis of this connection has been obscure. Studies attempting to link DNA repair deficits to neurobehavioral deficits have been hindered by the lack of models that faithfully recapitulate human disease. In the current study, we explored how deletion of the DNA repair protein histone H2AX affects motor learning and coordination.

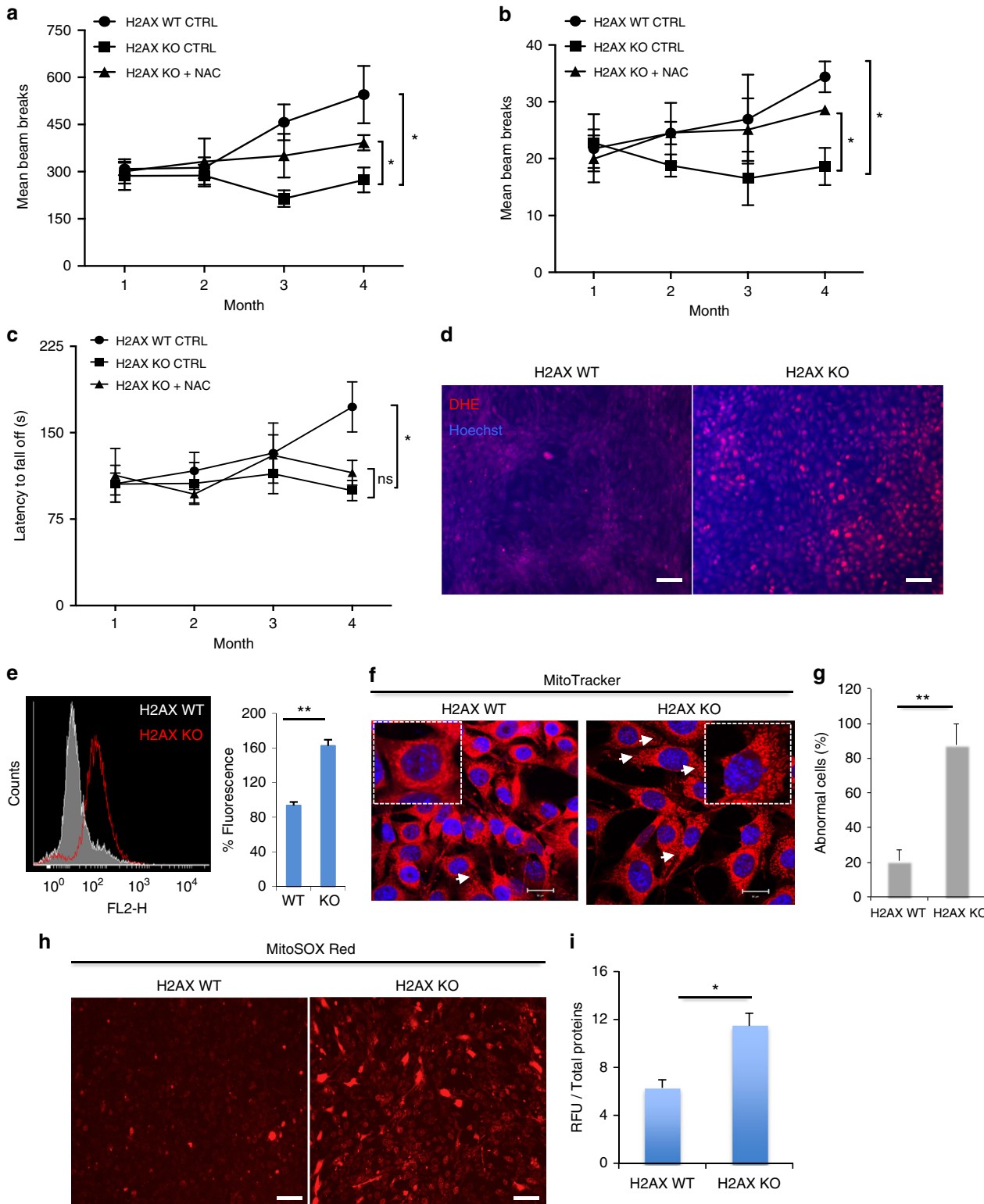

For many years, the histone H2AX has been primarily seen as a gene of DNA repair[6]. We showed that histone H2AX regulates motor behavior, as well as responses to oxidative stress. Utilizing mice deleted for H2AX, we defined functions for this histone in motor activity and balance using motor tests such as rotarod and open field. Our findings indicate that mice deficient for H2AX exhibit reduced general locomotor activity and impaired motor balance. Mitigation of ROS with the generic antioxidant NAC reversed some of the behavioral deficits. This beneficial effect of the NAC implies defects in the proper response to oxidant-mediated damage in the mutant mice. To identify specific genes associated with oxidative metabolism that may be altered in H2AX mutants, we conducted a genome-wide screen of genes involved in ROS metabolism and found that the major antioxidant gene NRF2 and its transcriptional target NQO1 were particularly diminished in cells lacking H2AX. This finding suggests that H2AX helps maintain redox balance by regulating specific transcriptional targets. Recently, Gruosso et al. described a role for ROS-mediated regulation of H2AX turnover in cancer therapy[27]. It is accepted that the accumulation of ROS leads to increased genomic instability, facilitating response to chemotherapeutic agents[34, 35]. Gruosso et al. also demonstrated that chronic oxidative stress enhances interactions of H2AX with the E3 ubiquitin ligase RNF168, which in turn fosters H2AX protein degradation by the proteasome. These observations are relevant to persistent oxidative stress conditions such as aging and aggressive breast cancers[27]. Our finding that H2AX deficiency per se elicits excessive ROS may reflect a vicious cycle in which chronic oxidative stress leads to H2AX protein degradation, which in turn creates a deleterious environment of enhanced oxidative lesions. The impaired antioxidant response is particularly relevant in the striatum, a region in the brain that is highly sensitive to ROS-mediated damage[26] and mediates motor learning[26, 36, 37]. Our findings that antioxidant treatment mitigates some of the neurobehavioral deficits and the ROS-mediated damages in the striatum, point to a role for H2AX-driven redox homeostasis in motor learning. On the other hand, treatment with the NAC did not seem to prevent the disturbed motor coordination observed in H2AX knockout mice evaluated in rotarod tests. Rotarod tests in mice have been widely used to illustrate impairment of motor coordination[11, 38].

In summary, the present study demonstrates that H2AX, a histone and classic DNA repair protein, influences the brain and behavior. H2AX participates in the maintenance of redox balance in cells and it does this by regulating specific transcriptional targets. Heretofore H2AX has been studied primarily as an agent of DNA repair. The present study extends the role of this protein to the arena of oxidative stress, in which we demonstrate a major influence of H2AX. H2AX seems to mimic ATM's role in redox homeostasis[13, 14], but unlike ATM which directly senses ROS, H2AX regulates several components of the antioxidant response. One of the principal risk factors for several age-related neurological disorders is the concomitant dysfunction of mitochondrial activity which results in increased ROS and decreased capacity of the DNA repair machinery[39, 40]. Our findings that deficiency of the DNA repair histone H2AX leads to accumulation of mitochondrial-derived ROS and neurobehavioral deficits imply a key role for H2AX in aging and neurodegenerative diseases.

## Methods

**Cell culture.** Mouse embryonic fibroblasts from both wild-type and H2AX mutant mice were obtained from the Developmental Therapeutics Branch, Laboratory of Molecular Pharmacology, National Cancer Institute, and National Institutes of Health. Cells were grown at 37 °C with 5% $CO_2$ in DMEM (Invitrogen, Carlsbad, USA), supplemented with 10% Fetal Bovine Serum (FBS, Atlanta Biologicals, GA, USA). All media were supplemented with 2 mM glutamine, penicillin, and streptomycin (Invitrogen, Carlsbad, CA, USA).

**Animals.** Mice were housed on a 12 h light-dark schedule and received food and water ad libitum. Dr. Nussenzweig (NCI/NIH) kindly provided the H2AX heterozygote mice. All animals were treated in accordance with the recommendations of the National Institutes of Health and approved by the Johns Hopkins University Committee on Animal Care. H2AX knockout mice offspring received 20 mM NAC in the drinking water starting from weaning.

**Motor tests.** General locomotor activity in H2AX knockout mice was assessed by open field analysis (Photobeam Activity System; San Diego Instruments). Mice were placed in the center of an enclosed acrylic chamber and allowed to explore freely for 45 min. Photobeams tracked horizontal movements and rearing behavior in the $x$ and $y$ directions. Following open field studies, motor learning, and coordination were examined in H2AX-null mice by the rotarod test (Rotamex-5; Columbus Instruments). Testing was conducted over 3 days with 2 days of training trials. Each mouse was then given three to four trials and latency to fall off the rotarod was measured as it accelerated from 4 to 40 rpm over a 5 min period. Before the start of testing on the first 2 days, each mouse was given a habituation trial by being placed on the rotarod, which was rotating at a constant speed of 4 rpm for 10 min.

**Generation of cells expressing H2AX-WT (H2AX-rescued cells).** For retroviral infection, HEK 293T cells were used for virus packaging according to the manufacturer's instructions. Briefly, retroviral constructs p-BABE-puro or pBABE-puro-H2AX-WT or pBABE-puro-H2AX-S139A, pVPack-VSVG and pVPack-GP were transfected into HEK 293T cells using Lipofectamine 2000 according to the manufacturer's instructions. Viral particles were harvested at 48 h post-transfection. Cells were infected with virus for 48 h in the presence of DEAE-dextran (10 μg ml$^{-1}$). Infected cells were either harvested for gene and protein expression analysis or selected to establish stable expression. For the plasmid pBABE-puro-H2AX construct, H2AX was amplified between *Bam*HI and *Eco*RI sites from pCR2.1 vector using the following primers: forward primer: 5′-GTCGGATCCATGTCGGGCCGCGG-3′ and reverse primer: 5′-

**Fig. 2** Oxidative stress is involved in H2AX loss-induced neurobehavioral deficits. **a, b** N-acetyl-cysteine (NAC) treatment prevents disturbed general motor activity. Open-field testing demonstrated that N-acetyl-cysteine (NAC 20 mM) significantly reduces impaired motor activity in H2AX knockout mice as measured by horizontal activity (**a**) or vertical activity (**b**), $y$-axis, number of beam breaks, $x$-axis, month. Data indicate the average of locomotor activity over 45 min each month. $n = 5$ (means ± s.e.m.) for H2AX wild type control group, $n = 7$ (means ± s.e.m.) for H2AX knockout control group, and $n = 8$ (means ± s.e.m.) for H2AX knockout treated with NAC. *$P < 0.05$. **c** Rotarod tests were also performed on mice used in **a, b**. Testing was conducted each month from weaning to 4 months, where each mouse was given three rounds of training, and latency to fall off the rotarod was measured at it accelerated from 4 to 40 rpm over 5 min; $y$-axis, the latency to fall off in seconds, $x$-axis, month. $n = 5$ (means ± s.e.m.) for H2AX wild type control group, $n = 7$ (means ± s.e.m.) for H2AX knockout control group, and $n = 8$ (means ± s.e.m.) for H2AX knockout treated with NAC. *$P < 0.05$, ns non-significant ($P = 0.1482$). **d, e** H2AX deletion leads to accrued ROS levels. Control cells (H2AX WT), as well as mutant mouse embryonic fibroblasts (H2AX KO) were incubated with dihydroethidium (DHE) and used for ROS detection by microscopy. Cells were counterstained using Hoechst (**d**). The raw DHE fluorescence was analyzed by flow cytometry (**e**), (Left, raw DHE fluorescence; right, quantification of the fluorescence mean). Data are means ± s.d.; $n = 3$. **$P < 0.001$. **f, g** Mitochondrial morphology was analyzed in both wild-type and H2AX mutants. Cells were stained with 500 nM MitoTracker and visualized on a Zeiss LSM 700 confocal laser scanning microscope; **f** representative image. When stained with the mitochondrial marker Mitotracker, the mitochondria in the H2AX KO cells appeared swollen and less filamentous than those in the control cells. The percentage of cells with these abnormalities is estimated in **g** and data are means ± s.d.; $n = 3$. **$P < 0.001$. **h** Representative images obtained by fluorescence microscopy of MitoSOX Red staining in control and H2AX mutant cells. Fluorescence intensity was quantified using Plate Reader (**i**), and data are means ± s.d.; $n = 3$, *$P < 0.05$. Scale bars: **d** and **h**, 100 μm. For all data, statistical significance was determined by a two-tailed, unpaired Student's $t$-test

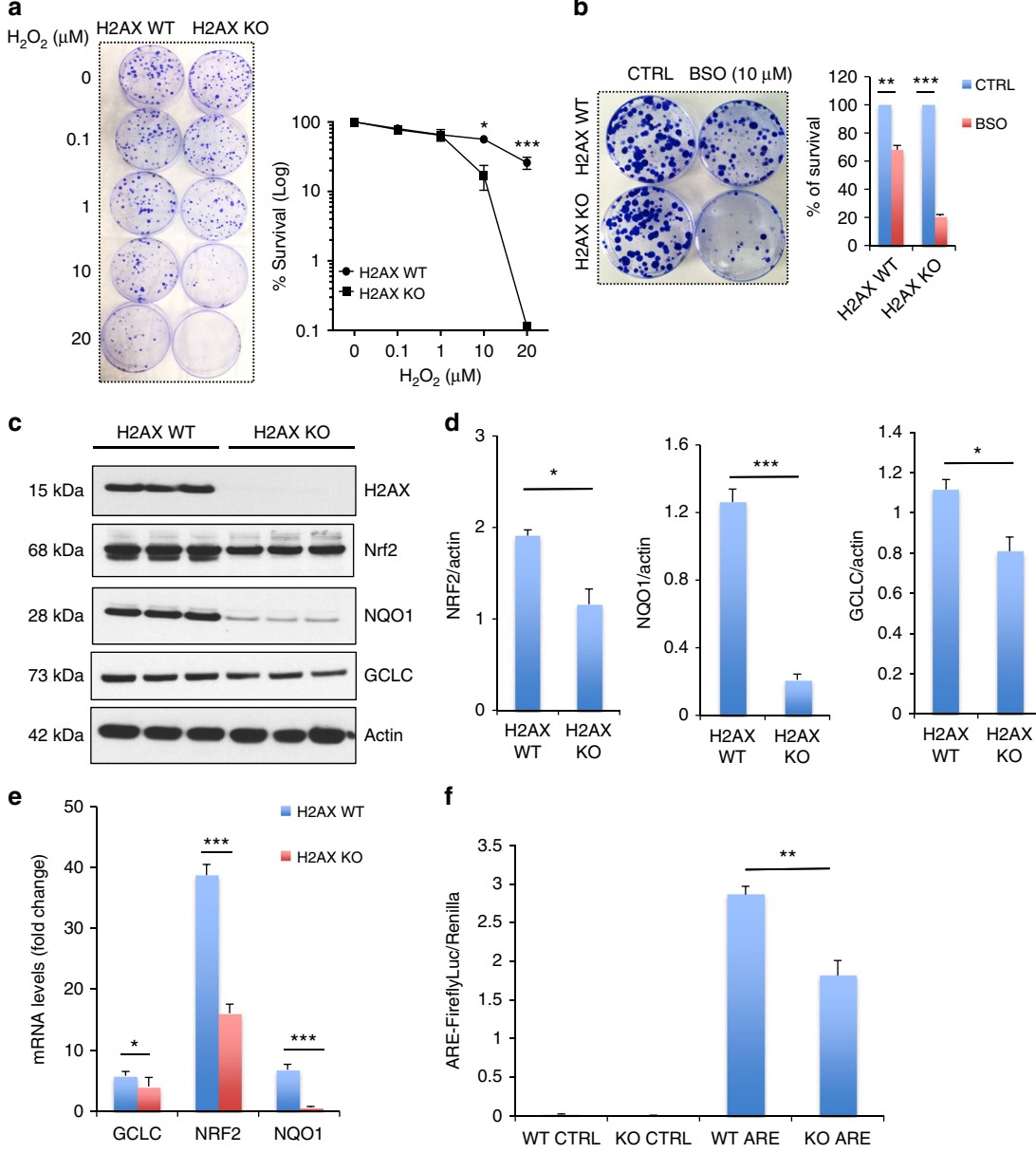

**Fig. 3** H2AX deficiency results in impaired NRF2-regulated antioxidant response. **a**, **b** Oxidants treatment elicits lower survival in mutant cells. Wild-type and H2AX knockout cells were treated with hydrogen peroxide ($H_2O_2$) (**a**) or 10 µM buthionine sulfoximine (BSO) (**b**), and cultured at 37 °C for 10 days. Colonies stained with Coomassie blue were counted for survival estimation. Data are means ± s.d.; $n = 3$. Statistical significance was determined by a two-tailed, unpaired Student's $t$-test. **$P < 0.001$, ***$P < 0.0001$. **c**, **d** H2AX deletion reduces the major antioxidant response protein NRF2 and its transcriptional targets GCLC and NQO1. Protein lysates from wild-type and H2AX knockout MEFs were processed for immunoblot detection of H2AX, NRF2, NQO1, GCLC. Actin was used as loading control. **c** Representative image and **d** right, quantification. Statistical significance was determined by a two-tailed, unpaired Student's $t$-test. Data are mean ± s.d.; $n = 3$, *$P < 0.05$, **$P < 0.001$. **e** Analysis of NRF2, NQO1, and GCLC transcript levels in wild-type and H2AX KO MEFs by real-time PCR. Expression values are relative fold change for gene transcripts normalized to GAPDH. Error bars represent s.e.m. ($n = 3$), *$P < 0.05$, ***$P < 0.0001$. **f** H2AX depletion reduces the promoter activity of the antioxidant response element (ARE) binding site for NRF2. Promoter activity was accessed by luciferase reporter assay in both control cells (WT) and H2AX mutant cells (KO). Error bars represent the s.e.m. ($n = 3$). Statistical significance was determined by a two-tailed, unpaired Student's $t$-test. **$P < 0.001$

GTAGAATTCTTAGTACTCCTGGGAGGCCTGG-3′. The PCR product was digested and subcloned into p-BABE-puro vector.

**Luciferase assay**. The promoter luciferase reporter assay was performed as previously described[5]. Cells were seeded in triplicate into a 24-well plate and cultured for 24 h. The promoter reporter plasmid for ARE, which is the binding site for NRF2, and control plasmid were transfected into the cells using Lipofectamine 2000 reagent (Life Technologies; Grand Island, NY, USA). Luciferase activity was measured 24 h after transfection following the manufacturer's instructions.

**Real-time PCR**. Total RNA was extracted from cells using RNeasy Mini Kit (Qiagen, Valencia, CA, USA) according to the manufacturer's instructions. Quality of RNA preparation, based on the 28S/18S ribosomal RNAs ratio, was assessed using the RNA 6000 Nano Lab-On-chip (Agilent Technologies, Palo Alto, CA, USA). Reverse transcription and RT-PCR were performed as previously described[41]. Oligonucleotides were pre-designed, validated and were considered to be proprietary information by Thermo Fisher Scientific. However, the assay IDs are available and are referenced as follow: NRF2 (Mm00477784_m1), NQO1 (Mm01253561_m1), GCLC (Mm00802655_m1).

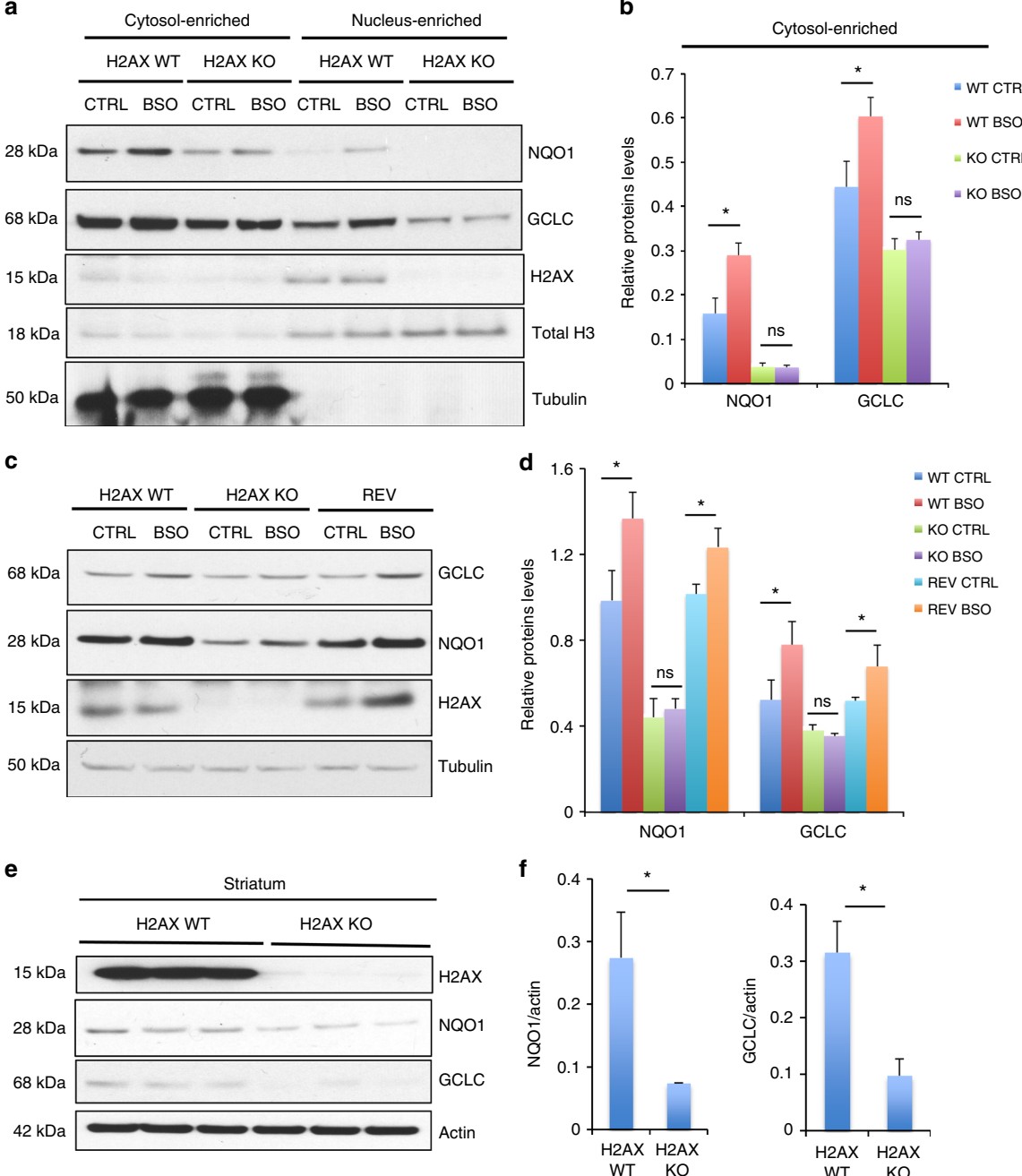

**Fig. 4** H2AX deficiency impairs re-expression of NRF2-transcriptional targets GCLC and NQO1 in response to oxidative stress. **a**, **b** BSO treatment results in impaired activation of NQO1 and GCLC expression in H2AX mutant cells. Immunoblot analysis of GCLC, NQO1, and H2AX in parental cells (H2AX WT) and H2AX knockout cells (H2AX KO) treated for 24 h with 2 mM buthionine sulfoximine (BSO) to induce endogenous oxidative stress. Both nuclear and cytosolic fractions were analyzed to gain insights into any differential changes in NRF2-transcriptional target expression. Data indicate that mutant cells failed to restore NQO1 and GCLC in response to oxidative stress in both cytosolic and nuclear fractions; **a** representative image and **b** quantification of the cytosolic fractions of NQO1 and GCLC. Similar trend was observed in the nuclear fractions. Error bars represent the s.e.m. ($n = 3$). Statistical significance was determined by a two-tailed, unpaired Student's $t$-test. *$P < 0.05$, ns non-significant ($P = 0.8164$ for NQO1 and 0.2783 for GCLC). **c**, **d** Immunoblot analysis of GCLC, NQO1, and H2AX in parental cells (H2AX WT), H2AX knockout cells (H2AX KO), and H2AX knockout cells in which H2AX expression was restored (Rescued), utilizing actin as a loading control. Cells were treated for 24 h with 2 mM BSO to elicit endogenous oxidative stress. **c** Representative image, **d** quantification. Error bars represent the s.e.m. ($n = 3$). Statistical significance was determined by a two-tailed, unpaired Student's $t$-test. *$P < 0.05$, ns non-significant ($P = 0.5241$ for NQO1 and 0.2069 for GCLC). **e**, **f** Immunoblot analysis of GCLC, NQO1, and H2AX in the striatum of both wild-type and mutant mice. **e** Representative image, **f** quantification. Error bars represent the s.e.m. ($n = 3$). Statistical significance was determined by a two-tailed, unpaired Student's $t$-test. *$P < 0.05$

**Clonogenic assay**. Cell survival was assessed by colony formation assay. Cells were trypsinized and identical numbers of H2AX wild-type MEFs (H2AX WT) and H2AX knockout cells (H2AX KO) were plated on 35 mm dishes. Cells were treated with either hydrogen peroxide ($H_2O_2$) or with BSO. After 10 days of incubation, the colonies were fixed with methanol for 10 min, and then stained with Coomassie blue. Colonies with >50 cells were counted. Clonogenic survival curves were constructed from at least three independent experiments.

**Western blots**. Cells were washed twice with PBS, solubilized in denaturing sample buffer and then subjected to SDS-PAGE. Proteins were electrotransferred to 0.2 μm Protran BA 83 nitrocellulose sheets (Invitrogen, Carlsbad, CA, USA) for immunodetection with the following primary antibodies: H2AX (1:2000; ab20669, Abcam, Cambridge, MA, USA); NRF2 (1:2000; MAB3925, R&D system, Minneapolis, MN, USA); NQO1 (1:3000; ab34173, Abcam, Cambridge, MA, USA); GCLC (1:2000; ab53179, Abcam, Cambridge, MA, USA). Immune complexes were detected with horseradish peroxidase coupled anti-rabbit or anti-mouse IgG antibodies (Amersham™, GE Healthcare, Pittsburgh, PA, USA).

**ROS detection**. Intracellular ROS measurements were performed using DHE. Briefly, cells were harvested and resuspended in 500 μL HBSS medium containing 10 μM DHE. Cells were then incubated for 30 min at 37 °C and used for analysis by flow cytometry (BD Biosciences, San Jose, CA, USA).

**Immunofluorescence confocal microscopy**. Mouse embryonic fibroblasts were stained with 500 nM MitoTracker (Life Technologies; Grand Island, NY, USA) for 30 min. Cells were subsequently fixed with 3.7% paraformaldehyde for 15 min and permeabilized with 0.2% Triton X-100 for 5 min. The cells were washed three times with PBS, mounted with antifade reagent with DAPI, and visualized on a Zeiss LSM 700 confocal laser scanning microscope. MitoTracker fluorescent intensity was determined at 579-nm excitation and 599-nm emission.

Mitochondrial superoxide generation was assessed in live cells with MitoSOX Red (Life Technologies; Grand Island, NY, USA), a fluorogenic dye that is taken up by mitochondria, where it is readily oxidized by superoxide, but not by other ROS or reactive nitrogen species. Cells were loaded with 5 μM MitoSOX Red in phenol-free DMEM for 10 min at 37 °C. Cells were washed with warm buffer (HBSS). MitoSOX Red fluorescent intensity was determined at 510-nm excitation and 580-nm emission. Mean fluorescence was quantified using a microplate reader and total protein quantification was used to estimate the differential levels of ROS.

**Statistical analysis**. Statistical analyses were performed using GraphPad Prism 7 software (GraphPad Software) and Microsoft Excel 2010. Parametric data were analyzed using a two-tailed $t$-test. A value of $P < 0.05$ was considered statistically significant. Data are presented as mean ± s.d.

**Data availability**. For the microarray analysis, total RNAs from MEFs were extracted using RNeasy Mini Kit (Qiagen, Valencia, CA, USA) following manufacturer's instructions. All RNAs were QC-tested and used at LMT/Affymetrix Group, NCI-Frederick, MD, to perform Affymetrix GeneChip mouse Gene 2.0 ST array ($n = 3$ for each group). Affymetrix Expression Console (EC) was used to generate CHP files from CEL files. Then, the CEL files were loaded into Partek Genomics Suite software, version 6.6 for normalization, and differential expression analysis. The NCBI/GEO accession number is GSE102548; and data are available at http://www.ncbi.nlm.nih.gov/geo/query/acc.cgi?acc = GSE102548. A one-way ANOVA was used to find statistically significant differentially expressed genes. There were 1295 distinct differentially regulated genes (fold-change > 2, FDR < 0.05). Agglomerative hierarchical clustering (both genes and samples) was carried out in Partek using average linkage method and Euclidean dissimilarity measure for both rows and columns. A heatmap was generated summarizing differentially expressed genes involved in ROS metabolism.

All other remaining data are available within the Article and Supplementary Files, or available from the authors upon request.

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

## Acknowledgements

We thank Roxanne Barrow, Lynda Hester, and Lauren Albacarys of the Solomon H. Snyder Department of Neuroscience (JHU); and Nancy Wong of the Laboratory of Genome Integrity (NCI/NIH) for help with experiments. We are indebted to Deeya Bhattacharya of the Johns Hopkins University for her help with experiments. This work was supported by the National Institutes of Health (NIH) and USPHS grants MH18501 and DA000266.

## Author contributions

U.W., B.D.P., and S.H.S. designed experiments. U.W., A.M.S., and B.D.P. performed experiments and analyzed data. U.W., W.M.B., and P.J. performed microarray analyses. A.N. and W.M.B. provided animal models and edited the manuscript. U.W. and S.H.S. wrote the manuscript.

## Additional information

**Competing interests:** The authors declare no competing interests.

