## [Peer Review File · Nature Communications]

Reviewers' comments:

Reviewer #1 (Remarks to the Author):

In this manuscript, Weyemi et al., reports that the deletion of H2AX in mouse brain leads to deficits in neurobehavioral responses, especially for motor function. In addition, the authors demonstrate that H2AX is involved in oxidative stress responses, which might explain the abnormality of neurobehavioral response in H2AX mutant mice.

Overall, this study might highlight a role of H2AX in the regulation of motor function via oxidative stress responses. But the lack of in vivo data for molecular mechanism, and methodological issues make the results highly questionable.

Major concerns:

1) Lack of histological and biochemistry data to support deficits in motor function in H2AX mutant mice.

Previous study in Atm mutant mice indicates that neuronal cell cycle reentry and degeneration (e.g. Purkinje cells) contribute to deficits in motor function (Li, et al 2012). The author should examine if there is any event of neuronal cell cycle reentry (e.g. cyclin A and cyclin D1) and degeneration (cleaved Cas-3) in H2AX mouse brain.

2) Lack of oxidative stress responses in primary neurons.

The evidence from MEFs and cell lines cannot fully account for oxidative stress responses in brain. The assays of oxidative stress responses need to be performed in primary neurons and microglia from wild type and mutant mice.

Minor concerns:

1) Figure 2 d, f, g:

The label bar should be added to demonstrate amplitude.

2) Figure 3 a, b:

The colorful image will be better.

3) Figure 4 a, b, c:

The same loading control should be better.

Reviewer #2 (Remarks to the Author):

The manuscript by Weyemi et al. examines a neural role of the Histone H2A variant, H2AX, by detailing the motor behavioral deficits in a H2AX knockout mouse model. Using motor behavioral assays (rota-rod and open-field) and cell-culture studies using mouse embryonic fibroblasts (MEFs) derived from H2AX-KO mice, the authors present evidence that deficient signaling interactions between H2AX and the transcription factor Nrf2n in the KO mice results in enhanced oxidative stress due to an impaired antioxidant response and may underlie the observed motor dysfunctions. The authors present data of an age-dependent impairment of rota-rod performance and deficits in general locomotor activity in the open-field in the H2AX-KO mice compared to WT controls. Furthermore, they suggest that these

motor deficits are mediated to some extent by increased oxidative stress by partially rescuing the open-field locomotor deficits with administration of the antioxidant N-acetylcysteine (NAC, 20mM) from the time of weaning to 4 months of age. To directly assess oxidative stress in H2AX-deficient cells, the authors generated MEFs from KO and WT mice and measured the increased accumulation of reactive oxygen species (ROS) with dihydroethidium staining as well as described abnormal mitochondrial morphology and levels of mitochondrial ROS in the H2AX-KO cells. Furthermore, the authors show that the H2AX-KO cells are especially vulnerable to oxidative damage from H₂O₂ and buthionine sulfoximine. Microarray analysis of the H2AX-KO mice revealed altered gene-expression in several genes involved in ROS metabolism but in particular, Nrf2 and its two major transcriptional targets, NQO1 and GCLC, were significantly down-regulated in the KO mice. Decreased activity of Nrf1 and downstream expression of NQO1 and GCLC was validated by RT-qPCR, Western blots, and promoter luciferase assays and was rescued in the H2AX-KO cells by viral-mediated re-expression of H2AX. Finally, to make the connection between the deficits in H2AX-Nrf1-mediated signaling of oxidative stress response and the motor deficits in the KO mice, the authors present evidence that the downstream signaling targets of Nrf1 (NQO1 and GCLC) are significantly reduced in the striatum of the H2AX-KO mice.

Overall, the strongest points of this study were the age-dependent motor impairment in H2AX-KO mice, and the cell-culture experiments with the H2AX-KO-derived MEFs which demonstrated the enhanced oxidative stress in the KO cells was due to diminished H2AX-Nrf1-mediated signaling of the antioxidant response. The study needs to be strengthened with new data to show elevated oxidative stress in the H2AX-KO mice at the age with motor deficits, and that the correction of motor phenotypes with NAC treatment is associated with reduction of oxidative stress in the brain, and not in the peripheral tissues.

Major Points:

- 1) The way the rotarod data presented in Figure 1 is not clear. The use of the terms "trials," "sessions," and "periods" is not consistent between the text, figure legend, and methods section. In Figure 1, are the authors presenting the average of the last session (of three) of each mouse for each Trial or are they averaging the 3 sessions for each mouse and averaging that for the Trial? Also, why are there 3 Trials for 8 weeks and 4 Trials for the 5-month data? There needs to be more consistency in describing and presenting this data. Similarly, Figure 2 only goes up to 4-months of age but not 5 months of age as in Figure 1; why did they make this change?
- 2) The authors did not indicate if the 8-week and 5-month data is from the same group of mice with motor behavior tracked across these two ages (repeat testing). At least with the rota-rod data, this needs to be known as it may alter interpretation of the data.
- 3) The number of mice per genotype used in all behavioral studies should be described in the figure legends.
- 4) The most critical deficiency in the data presented in this manuscript is the lack of in vivo data connecting the H2AX-KO to specific pathophysiology that can explain the impaired motor behavior. Although the authors indicate that the striatum of the H2AX-KO mice may

show some of the transcriptional deficiencies of NQO1 and GCLC (Figure 4) as seen in the MEFs, there is no mention of any neuropathological or biochemical studies done on the KO brains to show evidence of elevated oxidative stress in the aged H2AX-KO mice. Such a concern can be addressed by the addition of immunostaining or biochemical experiments to demonstrate increased markers of oxidative stress in the H2AX-KO brains, and immunostaining to show decreased expression of Nrf1, NQO1, or GCLC in brain regions including striatum.

5) The authors make a claim of abnormal mitochondrial morphology but present very little to validate this. There is no quantification of the MitoTracker staining (e.g., mitochondrial shape, size, or elongation) and even the image presented is too low-power to clearly emphasize the point of morphological differences.

6) The microarray data seems underpowered (n=3 / genotype) and insufficiently described. Age, sex-matching, brain regions assessed are not described and there are no statistics on the differential expression. Could they supply a supplementary table with statistics on the differentially expressed genes shown in Supplemental Figure 1?

7) The title of the manuscript is inaccurate. At best, the evidence provided implies that the motor deficits in the H2AX-KO mice are a consequence of ROS-mediated brain pathology, but that is a long way from being able to claim that "H2AX controls motor learning". They need to either modify this title, or provide direct evidence that genetic alteration of the levels of Nrf2 or H2AX in a specific brain region involved in motor learning (striatum) can modify the motor learning phenotypes in H2AX-KO mice.

Minor Points:

1) The representative image in Figure 2G over-exaggerates the quantified data in the same panel.

2) Westerns in Figure 4a and 4b need to be quantified.

Reviewer #3 (Remarks to the Author):

The manuscript by Weyemi et al. report a role for the histone variant H2AX in motor learning and neurological functions. H2AX KO mice are shown to exhibit several neurobehavioral phenotypes, which are interestingly rescued by reactive oxygen scavengers. The ROS pathway and response to ROS including H₂O₂ is shown to be defective in H2AX KO mice. These findings for the first time link the DNA repair histone variant H2AX to ROS pathways and neurological functions. This is a potentially very interesting study given that very little is known about the role of H2AX in these processes and more broadly, how DNA repair genes can influence ROS and neurological functions. This certainly is an interesting area of research that would be broadly interesting for the readers of Nature Communications. Currently however, the manuscript is somewhat underdeveloped as it does not demonstrate that it is the DNA repair function of H2AX that is involved. There

are also a few other issues that need to be addressed before publication.

Major Issues:

1. ATM is known to sense ROS and be involved in neurological processes. Given that H2AX is phosphorylated on S139 by ATM, it should be tested whether or not a S139A mutant of H2AX rescues or is deficient for ROS function in H2AX KO cells. This result would provide important mechanistic insights into the function of H2AX in these processes, which would vastly improve the impact of these findings.
2. In Figure 2A-C, there is no WT mice treated with NAC. This control has probably been done in other studies and shown to have no effect but a citation should at least be added to support this. If this information is not known, it would be difficult to compare these results without this additional treatment group.
3. In Figure 2D, does ATM inhibition act additively, synergistically or epistatically with H2AX loss. Given that ATM inhibitors are validated and available, this would be an easy but very informative experiment to perform. This relates to issue 1, which is an important idea that needs to be tested and developed for this study.
4. The data for figure 4 a and b are not very striking. To further support these findings, transcriptional analysis of these genes under these conditions should be performed to see whether or not these effects are due to transcriptional changes and to also further support the conclusions made by the authors based on these data.
5. The microarray datasets are very interesting but are not properly analyzed. Additional analysis should be performed given that 1295 genes are differentially regulated in WT versus H2AX KO cells. At the very least, GO Analysis should be performed to see which biological pathways are affected by the loss of H2AX. A better analysis of these data would be very useful for this study.

Minor Issues:

1. The discussion is rather limited in scope. The discussion should be extended to put the results of this paper into context for DNA repair and ROS. For example, a more comprehensive description of the Grusso et al. paper is warranted, especially given that these results are related to this work and expand these concepts to therapeutic treatments. It has also recently been shown that ATM senses ROS to regulate the transcription of several genes, including cytokines involved in cancer processes including cell migration and invasion (Chen et al. *Elife* 2015, PMID: 26030852). These and other studies could provide a more balanced and comprehensive discussion by which to compare and contrast the findings from this work with the literature.

Answers to reviewers' comments:

Reviewer #1 (Remarks to the Author):

In this manuscript, Weyemi et al., reports that the deletion of H2AX in mouse brain leads to deficits in neurobehavioral responses, especially for motor function. In addition, the authors demonstrate that H2AX is involved in oxidative stress responses, which might explain the abnormality of neurobehavioral response in H2AX mutant mice.

Overall, this study might highlight a role of H2AX in the regulation of motor function via oxidative stress responses. But the lack of in vivo data for molecular mechanism, and methodological issues make the results highly questionable.

We would like to thank the reviewer for his/her comments which we believe have helped to substantially improve the first version of the manuscript. These points have been addressed and discussed below. Additional experiments have helped to provide compelling evidence of enhanced oxidative stress in the brains of mice lacking H2AX. Changes in the main text are highlighted in bold allowing easier reading of the revised manuscript.

Major concerns:

1) Lack of histological and biochemistry data to support deficits in motor function in H2AX mutant mice. Previous study in Atm mutant mice indicates that neuronal cell cycle reentry and degeneration (e.g. Purkinje cells) contribute to deficits in motor function (Li, et al 2012). The author should examine if there is any event of neuronal cell cycle reentry (e.g. cyclin A and cyclin D1) and degeneration (cleaved Cas-3) in H2AX mouse brain.

[redacted]

[redacted]

2) Lack of oxidative stress responses in primary neurons.

The evidence from MEFs and cell lines cannot fully account for oxidative stress responses in brain. The assays of oxidative stress responses need to be performed in primary neurons and microglia from wild type and mutant mice.

We would like to thank the reviewer for raising this critical point. We have now assessed ROS-mediated damages in both primary neurons and brain tissue from mice. We have assayed for protein carbonylation to analyze the global protein oxidation levels as a hallmark of ROS-mediated proteins damage . As shown in Supplemental figure 5, H2AX deletion leads to 30% increase of protein oxidation levels in the striatum, and more than a two-fold increase in the primary neurons. These lesions are partially mitigated by NAC, consistent with our findings that some of the neurobehavioral deficits are corrected by antioxidant treatment. Interestingly, the elevated striatal protein damage paralleled our observations in H2AX-deficient mice of abnormal hindlimb clasp and clenching, characteristics which are reminiscent of mouse models with oxidative damage in areas such as the striatum (Supplemental Fig. 5e) (Paul BD et al., *Nature*. ;509:96-100 (2014)).

It is important to mention that the H2AX mouse model does not show signs of neurodegeneration at least at the age the experiments were performed (up to 5 months).

Minor concerns:

1) Figure 2 d, f, g:

The label bar should be added to demonstrate amplitude.

2) Figure 3 a, b:

The colorful image will be better.

3) Figure 4 a, b, c:

The same loading control should be better.

All these points have been addressed in the revised manuscript. An immunoblot of tubulin was added to figure 4a to show enrichment for cytosolic fractions.

Reviewer #2 (Remarks to the Author):

The manuscript by Weyemi et al. examines a neural role of the Histone H2A variant, H2AX, by detailing the motor behavioral deficits in a H2AX knockout mouse model. Using motor behavioral assays (rota-rod and open-field) and cell-culture studies using mouse embryonic fibroblasts (MEFs) derived from H2AX-KO mice, the authors present evidence that deficient signaling interactions between H2AX and the transcription factor NRF2 in the KO mice results in enhanced oxidative stress due to an impaired antioxidant response and may underlie the observed motor dysfunctions. The authors present data of an age-dependent impairment of rota-rod performance and deficits in general locomotor activity in the open-field in the H2AX-KO mice compared to WT controls. Furthermore, they suggest that these motor deficits are mediated to some extent by increased oxidative stress by partially rescuing the open-field locomotor deficits with administration of the antioxidant N-acetylcysteine (NAC, 20mM) from the time of weaning to 4 months of age. To directly assess oxidative stress in H2AX-deficient cells, the authors generated MEFs from KO and WT mice and measured the increased accumulation of reactive oxygen species (ROS) with dihydroethidium staining as well as described abnormal mitochondrial morphology and levels of mitochondrial ROS in the H2AX-KO cells. Furthermore, the authors show that the H2AX-KO cells are especially vulnerable to oxidative damage from H₂O₂ and buthionine sulfoximine. Microarray analysis of the H2AX-KO mice revealed altered gene-expression in several genes involved in ROS metabolism but in particular, NRF2 and its two major transcriptional targets, NQO1 and GCLC, were significantly down-regulated in the KO mice. Decreased activity of Nrf1 and downstream expression of

NQO1 and GCLC was validated by RT-qPCR, Western blots, and promoter luciferase assays and was rescued in the H2AX-KO cells by viral-mediated re-expression of H2AX. Finally, to make the connection between the deficits in H2AX-Nrf1-mediated signaling of oxidative stress response and the motor deficits in the KO mice, the authors present evidence that the downstream signaling targets of Nrf1 (NQO1 and GCLC) are significantly reduced in the striatum of the H2AX-KO mice.

Overall, the strongest points of this study were the age-dependent motor impairment in H2AX-KO mice, and the cell-culture experiments with the H2AX-KO-derived MEFs which demonstrated the enhanced oxidative stress in the KO cells was due to diminished H2AX-Nrf1-mediated signaling of the antioxidant response. The study needs to be strengthened with new data to show elevated oxidative stress in the H2AX-KO mice at the age with motor deficits, and that the correction of motor phenotypes with NAC treatment is associated with reduction of oxidative stress in the brain, and not in the peripheral tissues.

We would like to thank the reviewer for his/her thoughtful evaluation of our manuscript. All the concerns raised are cogent and have been substantially addressed in the revised manuscript. Changes added to the main text are highlighted in bold to allow easier reading of the revised manuscript.

MajorPoints:

1) The way the rotarod data presented in Figure 1 is not clear. The use of the terms “trials,” “sessions,” and “periods” is not consistent between the text, figure legend, and methods section. In Figure 1, are the authors presenting the average of the last session (of three) of each mouse for each Trial or are they averaging the 3 sessions for each mouse and averaging that for the Trial? Also, why are there 3 Trials for 8 weeks and 4 Trials for the 5-month data? There needs to be more consistency in describing and presenting this data. Similarly, Figure 2 only goes up to 4-months of age but not 5 months of age as in Figure 1; why did they made this change?

In the current version of the manuscript, the nomenclature for the behavioral tests was revised for consistency. For the rotarod test, testing was conducted over a three day period. The first two days were considered as training trials for the mice. Then on the third day of the training, each mouse was given three to four independent trials in order to assess the trend of learning performance. Each trial value relates to the average of the latency to fall off (in sec) for the total number of animals used per group.

Four trials were shown at 5 months compared to 3 trials in younger animals. This figure well exemplifies the trend of the learning performance in older mice.

Figure 2 only goes up to 4-months of age because the learning performance remain the same between 4 and 5 months. Therefore treatment with NAC was kept only for 4 months.

2) The authors did not indicate if the 8-week and 5-month data is from the same group of mice with motor behavior tracked across these two ages (repeat testing). At least with the rota-rod data, this needs to be known as it may alter interpretation of the data.

The same mice used for the rotarod analyses were employed for the open field tests following the same time points. For instance, the mice used for behavioral tests at 8 weeks were followed up to 5 months. Either at 8 weeks or 5 months of age, mice were given training trials before the final experiment. The behavioral tests have been performed several times and data are significantly representative of independent tests. It is important to mention that behavioral tests with NAC treatment were performed on independent cohorts of mice, and similar motor learning trends were observed.

3) The number of mice per genotype used in all behavioral studies should be described in the figure legends.

We thank the reviewer for this point which is now addressed in the revised manuscript.

4) The most critical deficiency in the data presented in this manuscript is the lack of *in vivo* data connecting the H2AX-KO to specific pathophysiology that can explain the impaired motor behavior. Although the authors indicate that the striatum of the H2AX-KO mice may show some of the transcriptional deficiencies of NQO1 and GCLC (Figure 4) as seen in the MEFs, there is no mention of any neuropathological or biochemical studies done on the KO brains to show evidence of elevated oxidative stress in the aged H2AX-KO mice. Such a concern can be addressed by the addition of immunostaining or biochemical experiments to demonstrate increased markers of oxidative stress in the H2AX-KO brains, and immunostaining to show decreased expression of Nrf1, NQO1, or GCLC in brain regions including striatum.

We agree with the reviewer that evidence of increased oxidative *in vivo* would enhance the study. We have now assessed ROS-mediated damage in both primary neurons and intact mice. We analyzed the global protein carbonylation levels as a reflection of ROS-mediated protein damage in whole extracts of cells or brain tissues. As shown in the supplemental figure 5, H2AX deletion leads to a 30% increase in protein oxidation in the striatum and more than a 2 fold increase in the primary neurons. These lesions are partially mitigated by NAC and are consistent with our findings that some of the neurobehavioral deficits are corrected by antioxidant treatment. Interestingly, the elevated striatal protein damage paralleled our observations in H2AX-deficient mice of abnormal hindlimb claspings and clenching, characteristics which are reminiscent of mouse models with oxidative damage in areas such as the striatum (Supplemental Fig. 5e) (Paul BD et al., *Nature*. ;509:96-100 (2014)).

It is important to mention that no gross morphological changes were observed in brain sections from both wild type and H2AX knockout mice, at least at the age the experiments were performed (up to 5 months). These observations suggest that molecular and biochemical changes might more likely account for the neurobehavioral deficits observed.

5) The authors make a claim of abnormal mitochondrial morphology but present very little to validate this. There is no quantification of the MitoTracker staining (e.g., mitochondrial shape, size, or elongation) and even the image presented is too low-power to clearly emphasize the point of morphological differences.

When stained with the mitochondrial marker Mitotracker, the mitochondria in the H2AX KO cells appeared swollen and less filamentous than those in the control cells. The percentage of cells with these abnormalities is shown in the revised manuscript (Figure 2g).

6) The microarray data seems underpowered (n=3 / genotype) and insufficiently described. Age, sex-matching, brain regions assessed are not described and there are no statistics on the differential expression. Could they supply a supplementary table with statistics on the differentially expressed genes shown in Supplemental Figure 1?

The microarray was performed on mouse embryonic fibroblasts. We have provided in the revised manuscript the full data including statistics for all the differentially expressed genes involved in ROS metabolism. Further description is also provided in the legends. We also provided a supplementary table with statistics on the differentially expressed genes presented in the Supplemental Fig. 3 (Supplemental table 1). Additionally, we indicated in the *Methods* section the NCBI/GEO accession number GSE75444; and data will be publicly available to the readers for any potential utilization towards future investigation on H2AX deficiency and pathway analysis (<http://www.ncbi.nlm.nih.gov/geo/query/acc.cgi?acc=GSE102548>).

7) The title of the manuscript is inaccurate. At best, the evidence provided implies that the motor deficits in the H2AX-KO mice are a consequence of ROS-mediated brain pathology, but that is a long way from being able to claim that “H2AX controls motor learning”. They need to either modify this title, or provide direct evidence that genetic alteration of the levels of NRF2 or H2AX in a specific brain region involved in motor learning (striatum) can modify the motor learning phenotypes in H2AX-KO mice.

We have proposed a new title that more faithfully summarises our findings on the role of H2AX in neurobehavioral deficits and the influences of ROS in these deficits. We agree with the reviewer that this work has not fully explored the specific role of NRF2 in the mitigation of neurobehavioral deficits. The revised manuscript is entitled: **Histone H2AX deficiency causes neurobehavioral deficits and impaired redox homeostasis.**

Minor Points:

1) The representative image in Figure 2G over-exaggerates the quantified data in the same panel.

The quantification in the figure 2g (currently 2h) is the mean fluorescence of DHE obtained using quantification in a plate reader. This slight difference may result from the differential sensitivity of the two independent methods used (microscopy vs. spectroscopy).

2) Westerns in Figure 4a and 4b need to be quantified.

Quantification for these figures is now provided (Figure 4b and 4d).

Reviewer #3 (Remarks to the Author):

The manuscript by Weyemi et al. report a role for the histone variant H2AX in motor learning and neurological functions. H2AX KO mice are shown to exhibit several neurobehavioral phenotypes, which are interestingly rescued by reactive oxygen scavengers. The ROS pathway and response to ROS including H₂O₂ is shown to be defective in H2AX KO mice. These findings for the first time link the DNA repair histone variant H2AX to ROS pathways and neurological functions. This is a potentially very interesting study given that very little is known about the role of H2AX in these processes and more broadly, how DNA repair genes can influence ROS and neurological functions. This certainly is an interesting area of research that would be broadly interesting for the readers of Nature Communications. Currently however, the manuscript is somewhat underdeveloped as it does not demonstrate that it is the DNA repair function of H2AX that is involved. There are also a few other issues that need to be addressed before publication.

We would like to thank the reviewer for his/her positive feedback on our manuscript and especially for the constructive comments which we believe have led to a substantial improvement of our study. Changes added to the main text are highlighted in bold to allow smooth-reading of the revised manuscript.

Major Issues:

1. ATM is known to sense ROS and be involved in neurological processes. Given that H2AX is phosphorylated on S139 by ATM, it should be tested whether or not a S139A mutant of H2AX rescues or is deficient for ROS function in H2AX KO cells. This result would provide important mechanistic insights into the function of H2AX in these processes, which would vastly improve the impact of these findings.

As indicated by the reviewer, one intriguing question is whether the DNA repair-associated motif of H2AX, phosphorylatable Ser139 residue, is required for the ability to control redox homeostasis. To test this, we generated H2AX knockout cells expressing a mutant H2AX (H2AX S139A) that is unable to detect DNA damage. Our data indicate that the activation of NRF2 transcriptional targets NQO1 and GCLC is impaired in H2AX mutant revertants, just as in the H2AX-null cells as revealed in western blot and real time PCR. These results suggest that the DNA repair function of H2AX is most likely critical for its regulation of ROS metabolism. Data are now provided in the revised manuscript and discussed (Supplemental Fig. 4c-f).

2. In Figure 2A-C, there is no WT mice treated with NAC. This control has probably been done in other studies and shown to have no effect but a citation should at least be added to support this. If this information is not known, it would be difficult to compare these results without this additional treatment group.

As appropriately noted by the reviewer, we did not include data from WT mice treated with NAC in the first version of the manuscript. This information is now provided (Supplemental figure 1). We also included in the revised manuscript, evidence from previous studies confirming these data :“ **Wild-type mice treated with**

NAC did not show noticeable difference in behavior (Supplemental Fig. 1). These observations are in line with previous findings showing a lack of NAC effects on control mice when comparing several abnormalities induced by ATM deficiency including genomic instabilities¹⁶.

3. In Figure 2D, does ATM inhibition act additively, synergistically or epistatically with H2AX loss. Given that ATM inhibitors are validated and available, this would be an easy but very informative experiment to perform. This relates to issue 1, which is an important idea that needs to be tested and developed for this study.

This point raised by the reviewer is of great interest as it helps to provide not only a comparative analysis of the role of H2AX and ATM in ROS metabolism, but also addresses the functional relationship between ATM and its direct target (H2AX) in the DNA repair pathway. As shown in the supplemental figure 2, ATM specific inhibition with KU55933 leads to a two-fold increase in ROS production in wild type cells. H2AX deletion significantly potentiates this ROS production. It is also important to mention that ROS levels in H2AX-deficient cells are higher compared to cells with ATM inhibition alone. Nevertheless, as the inactivation of the two proteins were performed using different approaches (pharmacological vs. genetic), we have preferred not to make any conclusive remark. Data are now included in the revised manuscript: **“Considering the evidence that ATM phosphorylates H2AX in response to DNA damage, and that ATM deletion promotes accumulation of reactive oxygen species, we explored the concomitant inactivation of ATM and H2AX. These observations fit with the evidence that combined ATM/H2AX deficiency causes embryonic lethality and substantial genomic instability²⁸”.**

4. The data for figure 4 a and b are not very striking. To further support these findings, transcriptional analysis of these genes under these conditions should be performed to see whether or not these effects are due to transcriptional changes and to also further support the conclusions made by the authors based on these data.

We agree with the reviewer that this information is essential for a comprehensive mechanistic of H2AX loss-induced impairment of antioxidant response. Upon oxidative stress generated by BSO treatment, activation of the transcripts of NRF2-regulated genes GCLC and NQO1 is impaired in H2AX knockout cells (Supplemental Fig. 4a,b), further confirming the evidence described in the **figure 4f** that the promoter activity of NRF2-binding site (ARE) is reduced in H2AX-deficient cells.

5. The microarray datasets are very interesting but are not properly analyzed. Additional analysis should be performed given that 1295 genes are differentially regulated in WT versus H2AX KO cells. At the very least, GO Analysis should be performed to see which biological pathways are affected by the loss of H2AX. A better analysis of these data would be very useful for this study.

The referee is concerned that the microarray datasets are not comprehensively analyzed. This misapprehension reflects the lack of information regarding how the ROS genes list was generated. The heat map provided in the manuscript (currently supplemental Fig. 3) was generated by creating the list of genes involved in ROS metabolism from the 1295 differentially expressed genes (DEGs), on the basis of their annotation in Gene Ontology. This information is now explicitly indicated in the revised manuscript: **“To identify specific genes associated with oxidative metabolism that may be altered in H2AX mutants, we conducted a genome-wide differential gene expression analysis comparing H2AX-deficient and control mouse embryonic fibroblasts. appeared to be particularly relevant (Supplemental Fig. 3 and Supplemental Table 1)”**. We also provided a supplementary table with statistics on the differentially expressed genes presented in the

supplemental fig. 3 (Supplemental Table 1). Likewise, we indicated in the *Methods* section the NCBI/GEO accession number GSE75444; and data will be publically available to the readers for any potential utilization towards future investigation on H2AX deficiency and pathways analysis (<http://www.ncbi.nlm.nih.gov/geo/query/acc.cgi?acc=GSE102548>).

Minor Issues:

1. The discussion is rather limited in scope. The discussion should be extended to put the results of this paper into context for DNA repair and ROS. For example, a more comprehensive description of the Gruosso et al. paper is warranted, especially given that these results are related to this work and expand these concepts to therapeutic treatments. It has also recently been shown that ATM senses ROS to regulate the transcription of several genes, including cytokines involved in cancer processes including cell migration and invasion (Chen et al. *Elife* 2015, PMID: 26030852). These and other studies could provide a more balanced and comprehensive discussion by which to compare and contrast the findings from this work with the literature.

We agree with the reviewer for this comment. A more comprehensive and contextual discussion of the data in regard to the published studies mentioned above is now addressed in the revised manuscript.

REVIEWERS' COMMENTS:

Reviewer #1 (Remarks to the Author):

In the revised manuscript entitled "Histone H2AX deficiency causes neurobehavioral deficits and impaired redox homeostasis", the authors have carefully taken all of my comments into consideration. I appreciate the thorough effort to do a lot more work that enhances the manuscript. Particularly, the performance of histological and biochemical experiments to preclude PCs degeneration involved in deficits in motor function in H2AX mutant mice. In a sum, the authors have addressed my concerns and I think this manuscript is suitable for the general readership of Nature Communication.

Reviewer #2 (Remarks to the Author):

The authors have thoroughly revised their manuscript with the addition of new data. The revision has fully addressed all the issues raised by this reviewer. I do not have any further questions, and considered the manuscript suitable for publication in Nature Communication.

Reviewer #3 (Remarks to the Author):

The revised manuscript by Weyemi et al. provides additional new data that corroborate the involvement of H2AX and the DNA damage response in neurobehavioral responses and redox homeostasis. Specifically, new data is provided showing that the phosphorylated form of H2AX, S139, is required for these functions. An analysis with ATM is also performed, suggesting that H2AX and ATM partially function together in this pathway, although it is likely that ATM plays additional roles. Several new controls and textual changes have been added, including an extended discussion section that better puts this work in context of previous studies, which is a nice addition. Collectively, this revised manuscript is now ready for publication in Nature Communications.

REVIEWERS' COMMENTS:

Reviewer #1 (Remarks to the Author):

In the revised manuscript entitled “Histone H2AX deficiency causes neurobehavioral deficits and impaired redox homeostasis”, the authors have carefully taken all of my comments into consideration. I appreciate the thorough effort to do a lot more work that enhances the manuscript. Particularly, the performance of histological and biochemical experiments to preclude PCs degeneration involved in deficits in motor function in H2AX mutant mice. In a sum, the authors have addressed my concerns and I think this manuscript is suitable for the general readership of Nature Communication.

Reviewer #2 (Remarks to the Author):

The authors have thoroughly revised their manuscript with the addition of new data. The revision has fully addressed all the issues raised by this reviewer. I do not have any further questions, and considered the manuscript suitable for publication in Nature Communication.

Reviewer #3 (Remarks to the Author):

The revised manuscript by Weyemi et al. provides additional new data that corroborate the involvement of H2AX and the DNA damage response in neurobehavioral responses and redox homeostasis. Specifically, new data is provided showing that the phosphorylated form of H2AX, S139, is required for these functions. An analysis with ATM is also performed, suggesting that H2AX and ATM partially function together in this pathway, although it is likely that ATM plays additional roles. Several new controls and textual changes have been added, including an extended discussion section that better puts this work in context of previous studies, which is a nice addition. Collectively, this revised manuscript is now ready for publication in Nature Communications.

Our answer

We would like to thank all the reviewers for their positive feedback on our work.